# Emergency Department Visits in Children Associated with Exposure to Ambient PM_1_ within Several Hours

**DOI:** 10.3390/ijerph20064910

**Published:** 2023-03-10

**Authors:** Yachen Li, Lifeng Zhu, Yaqi Wang, Ziqing Tang, Yuqian Huang, Yixiang Wang, Jingjing Zhang, Yunquan Zhang

**Affiliations:** Hubei Province Key Laboratory of Occupational Hazard Identification and Control, Institute of Social Development and Health Management, School of Public Health, Wuhan University of Science and Technology, Wuhan 430065, China

**Keywords:** PM_1_, PM_2.5_, children, emergency department visits, intra-day exposure, case-crossover study

## Abstract

Background: Emerging evidence has integrated short-term exposure to PM_1_ with children’s morbidity and mortality. Nevertheless, most available studies have been conducted on a daily scale, ignoring the exposure variations over the span of a day. Objective: The main intention of this study was to examine the association between pediatric emergency department visits (PEDVs) and intra-day exposures to PM_1_ and PM_2.5_. We also aimed to investigate whether a high PM_1_/PM_2.5_ ratio elevated the risk of PEDVs independent from PM_2.5_ exposure within several hours. Methods: We collected hourly data on aerial PM_1_ and PM_2.5_ concentrations, all-cause PEDVs, and meteorological factors from two megacities (i.e., Guangzhou and Shenzhen) in southern China during 2015–2016. Time-stratified case-crossover design and conditional logistic regression analysis were used to assess the associations of PEDVs with exposures to PM_1_ and PM_2.5_ at different lag hours. The contribution of PM_1_ to PM_2.5_-associated risk was quantified by introducing PM_1_/PM_2.5_ ratio as an additional exposure indicator in the analysis adjusting for PM_2.5_. Subgroup analyses were performed stratified by sex, age, and season. Results: During this study period, 97,508 and 101,639 children were included from Guangzhou and Shenzhen, respectively. PM_1_ and PM_2.5_ exposures within several hours were both remarkably related to an increased risk of PEDVs. Risks for PEDVs increased by 3.9% (95% confidence interval [CI]: 2.7–5.0%) in Guangzhou and 3.2% (95% CI: 1.9–4.4%) in Shenzhen for each interquartile range (Guangzhou: 21.4 μg/m^3^, Shenzhen: 15.9 μg/m^3^) increase in PM_1_ at lag 0–3 h, respectively. A high PM_1_/PM_2.5_ ratio was substantially correlated with increased PEDVs, with an excess risk of 2.6% (95% CI: 1.2–4.0%) at lag 73–96 h in Guangzhou and 1.2% (95% CI: 0.4–2.0%) at lag 0–3 h in Shenzhen. Stratified analysis showed a clear seasonal pattern in PM-PEDVs relationships, with notably stronger risks in cold months (October to March of the following year) than in warm months (April to September). Conclusions: Exposures to ambient PM_1_ and PM_2.5_ within several hours were related to increased PEDVs. A high PM_1_/PM_2.5_ ratio may contribute an additional risk independent from the short-term impacts of PM_2.5_. These findings highlighted the significance of reducing PM_1_ in minimizing health risks due to PM_2.5_ exposure in children.

## 1. Introduction

Based on the 2019 Global State of the Air report, ambient atmospheric pollution is the fifth most dominant global risk factor for mortality, resulting millions of deaths annually from inhaling particulate matter (PM) [1]. Children are more susceptible to the hazardous influences of PM exposure, owing to their evolving defense mechanisms and the immaturity of their lungs [2]. Numerous epidemiological studies have associated short-term exposure to fine particulate matter in the atmosphere (PM_2.5_, aerodynamic diameter ≤ 2.5 µm) with a variety of pediatric injurious health outcomes [3,4]. However, prior analyses were mostly based on a daily timescale, thus failing to fully capture the risks of PM_2.5_ changes within a day.

During the past decade, emerging evidence emphasized elevated morbidity risks in children within a few hours after PM_2.5_ exposure [5]. Meanwhile, present academic interest in submicron particulate matter (PM_1_, aerodynamic diameter ≤ 1 µm) [6,7], a predominant constituent of PM_2.5_, is ongoing, and accumulating evidence indicated greater health hazards among children related to exposure to smaller particles [8]. Despite this, the health effects of sub-daily PM_1_ exposure on children remain largely unstudied due to scarce monitoring of ambient PM_1_ across the globe.

To fill these research gaps, we performed a case-crossover study in two southern cities in China using hourly records of PM measurements and children’s emergency department visits (EDVs). The primary purpose of the research was to inquire about the relationships between sub-daily exposures to PM_1_ and PM_2.5_ with pediatric EDVs (PEDVs). Secondarily, we also aimed to analyze whether a high PM_1_/PM_2.5_ ratio independently contributed to the increased risks of PEDVs.

## 2. Materials and Methods

### 2.1. Data Collection

We collected hourly PEDV records from two large hospitals in urban areas in Shenzhen (1 March 2015–31 December 2016) and Guangzhou (1 January 2015–31 December 2016), southern China. All-cause PEDVs were included in this study, with subcategories by sex (boys and girls) and age (0–4 years, 5–9 years, 10–14 years).

Hourly measurements of ambient particulate matter (PM_1_ and PM_2.5_) for 2015–2016 were gleaned from stationary meteorological sites in Shenzhen and Guangzhou, which are parts of the Chinese atmospheric monitoring network and supervised by the China Meteorological Administration [9]. Site-based PM statistics were monitored once each 5 min with quality-control procedures [10], using a GRIMM 180 Multi-channel Aerosol Spectrometer (Grimm Aerosol Technik GmbH, Ainring, Germany) [11,12]. Hourly average gaseous pollutants concentrations (nitrogen dioxide [NO_2_], sulfur dioxide [SO_2_], carbon monoxide [CO], and ozone [O_3_]) in Guangzhou and Shenzhen were gathered from the China National Urban Air Quality Real-time Publishing Platform (http://106.37.208.233:20035/, accessed on 24 November 2022). The global hourly meteorological dataset from the National Centers for Environmental Information (NCEI, https://www.ncei.noaa.gov/, accessed on 24 November 2022) was accessed to derive average temperature and relative humidity for Guangzhou and Shenzhen.

### 2.2. Statistical Analyses

#### 2.2.1. Main Analysis

We utilized the time-stratified case-crossover design (TSCC) separately in the two cities, which could effectively eliminate the confounding impacts of time-invariant individual factors (e.g., demographical, metabolic, and behavioral) in short-term studies. In the current study, the month and year were used as a defined time stratum, and the same hour on the same day of the week in the calendar month of a pediatric emergency visit was selected as the control time period. To control for the potential holiday effects on staffing and service delivery [13], we included a dichotomous variable in the model to indicate whether a particular date was a national public holiday. Conditional logistic regression (CLR) was applied to investigate connections of PEDVs with intra-day exposures to PM_1_ and PM_2.5_. In our main analytical model, we aligned the non-linear confounding effects of meteorological conditions using the natural cubic spline function (NCS) with three degrees of freedom (*df*) for the three-day moving average temperature and humidity terms [14,15].

PM_1_ and PM_2.5_ were separately analyzed as linear terms at different lag hours to estimate the odds ratios (ORs) and 95% confidence intervals (CIs) related to each interquartile range (IQR) rise in PM exposure, respectively. The contribution of PM_1_ to PM_2.5_-associated risk was quantified by introducing PM_1_/PM_2.5_ ratio as an additional exposure indicator in this analysis, adjusting for PM_2.5,_ and presented as the estimated ORs (95% CIs) per 10% increase in the proportion of PM_1_ in PM_2.5_. We used the NCS (*df* = 3) term smoothed exposure-response curves for PM_1_, PM_2.5,_ and PM_1_/PM_2.5_ ratio with trimmed observations at the 1st and 99th percentiles of the distribution and performed likelihood ratio tests to check the hypothesis of linear relationships by comparing the model fit of CLR models with linear or NCS terms for exposures.

#### 2.2.2. Subgroup Analyses

We conducted stratified analysis by sex (boys and girls) and age (0–4, 5–9, 10–14 years). To analyze seasonal differences in PM-PEDVs associations, the admission dates of cases were divided into cold and warm months. The warm months were defined as April to September, while the cold months ranged from October to March of the following year. We used meta-regression (MR) methods [16,17] to contrast variances in the effects of PM_1_, PM_2.5,_ and high PM_1_/PM_2.5_ ratio among subgroups (sex, age, and season). In the MR model, specifically, the dependent variables are stratified estimates of the impacts of PM_1_, PM_2.5_, and high PM_1_/PM_2.5_ ratio on PEDVs, while the meta-predictors are indicator variables for each stratum (sex, age, and season).

#### 2.2.3. Sensitivity Analyses

To inspect the robustness of our principal outcomes, we did certain sensitivity analyses by varying the modeling choices. We used bi- and tri-pollutant models to account for the effects of co-exposures to gaseous pollutants (e.g., NO_2_, SO_2_, CO, and O_3_). To avoid potential multicollinearity, we only included pollutants with Spearman’s correlation coefficients <0.7. In addition, we performed a variance inflation factor (VIF) analysis to test for the presence of multicollinearity in the multi-pollutant models [18]. Also, we changed the dfs from four to six for NCS function terms and varied the period of exposure for environmental temperature and relative humidity to check whether the impacts of meteorological factors were fully controlled in the main analysis.

Data were analyzed using R software (version 4.2.0, R Foundation for Statistical Computing, Vienna, Austria). The “mvmeta” package was utilized for MR analysis and the “survival” package for CLR modeling. All tests were two-sided, and the *p*-value <0.05 was considered statistically significant.

## 3. Results

Table 1 summarizes the basic characteristics of PEDVs during 2015–2016. A total of 97,508 and 101,639 cases were enrolled in Guangzhou and Shenzhen, respectively, equivalent to a mean of 5.6 (standard deviation [SD] = 5.5) and 6.3 (SD = 4.3) visits per hour. In both cities, boys accounted for over 60% of cases, and more than half of visits were children aged 0–4 years.

Table 2 outlines the cumulative distributions of meteorological conditions and environmental atmospheric pollutants during case and control periods. In both cities, slightly higher concentrations of PM_1_ and PM_2.5_ were noted in case hours than those in control hours (e.g., mean PM_1_: 26.9 µg/m^3^ versus 26.4 µg/m^3^ in Guangzhou; 17.9 µg/m^3^ versus 17.7 µg/m^3^ in Shenzhen). Similar patterns were identified for PM_1_/PM_2.5_ ratio in Shenzhen, exhibiting greater proportions of PM_1_ in PM_2.5_ in case hours (75.9% versus 75.7%). However, the proportions of PM_1_ and PM_2.5_ were identical on the case and control days in Guangzhou, which were both 84.5%. PM_2.5_ was highly and positively correlated with PM_1_ (r_Guangzhou_ = 0.99, r_Shenzhen_ = 0.96), but showed a weak correlation with PM_1_/PM_2.5_ (r_Guangzhou_ = 0.24, r_Shenzhen_ = 0.36) (Appendix A).

Figure 1 illustrates the assessed PM-related risks for PEDVs based on diverse lag patterns in Guangzhou and Shenzhen. Ambient exposure to PM_1_ and PM_2.5_ were notably correlated with an enhanced risk of PEDVs within several hours. Guangzhou and Shenzhen displayed discernible lag patterns in PM-PEDVs relationships, whereas these patterns were normally comparable for PM_1_ and PM_2.5_ in an identical city. In Guangzhou, OR estimates were marginally elevated from lag 0–3 to 4–6 h and were subsequently attenuated to the null after lag 25–48 h. In Shenzhen, PM_1_- and PM_2.5_-related risks peaked at lag 7–12 h and became insignificant after lag 49–72 h. High PM_1_/PM_2.5_-related risks in Guangzhou were only observed at lag 49–72 and 73–96 h, whereas significant effects appeared immediately at lag 0–3 h and lasted until lag 49–72 h in Shenzhen. Per 10% rise in PM_1_/PM_2.5_ ratio, for instance, risks at lag 49–72 h enhanced by 2.6% (OR = 1.026, 95% CI: 1.011–1.040) for Guangzhou (Appendix A) and 1.2% (OR = 1.012, 95% CI: 1.003–1.022) for Shenzhen (Appendix A), respectively. Positive correlations of PEDVs with PM_1_ and PM_2.5_ largely remained in our sensitivity analyses, and effect estimates of the relationships were broadly robust to the additional adjustment of ambient gaseous pollutants and modeling specifications of exposure periods and dfs for temperature and relative humidity (Appendix A).

Figure 2 manifests the shapes of exposure-response associations of PEDVs with PM_1_, PM_2.5_ and PM_1_/PM_2.5_ ratio. In Guangzhou, the curves for PM_1_ and PM_2.5_ showed a similar non-linear pattern with a steeper gradient at low concentrations, suggesting that children were probably more susceptible to the negative influences of atmospheric particulate matter at lower levels. However, approximate linear relationships (*p* < 0.05 for nonlinearity) were observed in curves for PM_1_ and PM_2.5_ in Shenzhen, indicating no significant threshold concentration below with no excess risk of PEDVs identified. The exposure-response curves for PM_1_/PM_2.5_ ratio in the two cities departed from linearity, with an increased risk of PEDVs with improving exposure, followed by a sharp elevation or reduction in risk after an estimated breakpoint of >85%.

Figure 3 presents estimated ORs of subgroups stratified by sex and age. In Guangzhou, exposures to PM_1_ and PM_2.5_ had remarkable impacts on both sex and all ages, suggesting no evidence for modifying the effects of sex and age. In Shenzhen, however, PM_1_- and PM_2.5_-related risks were only found among boys and children under ten years, and remarkable sex differences were identified (PM_1_, *p* = 0.028; PM_2.5_, *p* = 0.018). The relationship between the high PM_1_/PM_2.5_ ratio and PEDVs also differed in the two cities. High PM_1_/PM_2.5_-related risks were only observed among boys and under−five children only in Shenzhen but among both sexes and all age groups in Guangzhou. Specifically, with a 10% rise in the proportion of PM_1_ in PM_2.5_, the risk of PEDVs elevated by 1.7% (95% CI: 0.0–3.5%) at lag 49–72 h in Guangzhou and 1.4% (95% CI: 0.3–2.6%) at lag 0–3 h in Shenzhen for children under five years old.

Table 3 computes PM-related risks for PEDVs stratified by season. Significantly stronger associations with PM_1_ and PM_2.5_ were consistently observed in cold months in two cities, despite no excess risks occurring in warm months in Shenzhen. For example, PEDVs risks in Guangzhou increased by 6.8% (95% CI: 5.3–8.4%) in the cold season and 3.7% (95% CI: 2.1–5.2%) in the warm season for each IQR rise in PM_1_. A high PM_1_/PM_2.5_ ratio exhibited stronger impacts on PEDVs in cold months in Guangzhou, and evident warm–cold differences were only identified in Guangzhou (*p* = 0.025). For a 10% increase in PM_1_/PM_2.5_ ratio, the risk of PEDVs during the cold season was 1.044 (95% CI: 1.009–1.080) in Guangzhou at lag 49–72 h. A more detailed analysis of the seasonal effects across various lag hours can be found in Appendix A.

## 4. Discussions

To the best of our knowledge, this is the first study to quantify the contribution of intra-day PM_1_ exposure in the PM_2.5_-associated risk of PEDVs in the analysis. We utilized a case-crossover design to examine the relations between PM_1_, PM_2.5,_ and PM_1_/PM_2.5_ ratio and the risk of PEDVs at a sub-daily timescale. PM_1_ and PM_2.5_ exposures within a few hours were identified to be linked to enhanced risk of PEDVs, and a high PM_1_/PM_2.5_ ratio may play an independent role in triggering the risk of PEDVs. In the subgroup analysis, PM-PEDV associations were stronger in the cold months than in the warm months.

By performing a case-crossover analysis on approximately 0.2 million PEDVs, we discerned that temporary exposure to environmental PM_1_ and PM_2.5_ markedly enhanced the risks of PEDVs in the first few hours of exposure. To some extent, this intraday pattern concerning PM_2.5_ effects on children was generally echoed by several preceding studies in China [19], Japan [20,21], and Australia [22]. Nevertheless, the time window of susceptibility to health impacts of PM_2.5_ in children varied among these studies. For instance, PEDVs were found to be related to PM_2.5_ exposure at lag 0–1 h in 11 Japanese cities [20] but at lag 0 h in Shenzhen, China [23]. These studies highlighted that the first few hours might be a susceptible exposure time window, and current air quality standards based on 24-h average concentrations may be inadequate to protect children from intraday exposure to PM_2.5_ air pollution. Additionally, our study provided a comparative insight into short-term associations of PM_1_ and PM_2.5_ exposure in raising the risk of childhood illnesses within a day (Figure 1), which coincided with a daily time-series study on EDVs for all ages’ coverage of 26 cities across China [11]. Highly consistent effect patterns between PM_1_ and PM_2.5_ were also reported in previous children’s studies on the consequences of daily hospital admission and hospitalizations for cardiovascular [7,17] and respiratory diseases [2,6], along with EDVs [24,25]. When developing strategies to protect children’s health, local authorities should thus consider the adverse effects of sub-daily scale air pollutants to help children avoid exposure to high-risk time windows. Additionally, more extensive analysis is warranted to clearly distinguish the unfavorable impacts of PM_1_ and PM_2.5_ on children and highlights the urgency of developing air quality standards regarding ambient PM_1_ for the protection of children in the coming generations.

In the current study, we associated elevated risk of PEDVs with a high PM_1_/PM_2.5_ ratio independent from the effect of PM_2.5_ exposure, suggesting a predominant role of PM_1_ in PM_2.5_-related health risk in children. Generally, accordant findings were seen in a recent multi-city cross-sectional study linking a 4.0% (95% CI: 2.0–5.0%) excess risk of childhood asthma with per 1% increase in the ratio of early-life PM_1_ to PM_2.5_ [26]. Also, there is emerging time-series or case-crossover evidence in China showing that children’s health risks related to short-term PM_2.5_ exposure could be mostly attributed to PM_1_ [15,27]. Recent findings indicated that short-term exposure to ultrafine particulate matter (PM_0.1_, aerodynamic diameter ≤ 0.1 µm) was associated with a number of cardiovascular events [28,29], and several studies conducted in China further illustrated that the hazardous impacts of PM_0.1_ were independent of other size-segregated particles [30] and were stronger than larger particles [30,31]. The underlying biological mechanism is that PM_1_ or PM_0.1_ could be more likely to reach the alveolar portion of the respiratory tract, irritating the alveolar walls and promoting oxidative stress and inflammation, thereby impairing lung function [32,33]. Moreover, PM_1_ or PM_0.1_ features a higher surface-to-volume ratio compared to PM_2.5_, and thus more toxins from human emissions can be attached [8]. From the view of public health interventions, more exposure-health research is of great necessity to better understand the children’s health risks correlated with ambient exposure to submicron and ultrafine particulate matter, given the extensive lack of national routine monitoring.

In line with several prior investigations [34,35], we detected mixed evidence for effect modification by age and sex in two included cities. In season-stratified analysis, PM-associated risk of PEDVs appeared to be stronger in cold months, which was largely consistent with the findings of prior Chinese studies conducted in southern cities [15,23,36]. The underlying mechanisms for effect differences between warm and cold months remain unclarified, but seasonal variations could be possibly interpreted by pollution sources, chemical compositions, and climate conditions [37]. The cold-warm disparity highlighted the need for the government to take timely and targeted measures to further reduce peak season PM pollution in order to alleviate the related disease burden in children. In addition, personalized mitigation measures among children (particularly those with existing respiratory diseases) should also be strengthened, for example, by reducing time spent outside in the cold months or increasing intake of antioxidant polyunsaturated fatty acids and vitamins (e.g., more fish, fresh vegetables, and fruit) to mitigate the adverse effects of PM [8].

Some limitations of this research should be noted. First, because particulate matter data (PM_1_ and PM_2.5_) in Guangzhou and Shenzhen were obtained from fixed sites, it is possible to misclassify individual exposures and bias the estimated associations. Second, indoor air pollution in homes, kindergartens, or schools was not considered in this study. Third, the PEDVs data were obtained from two large conventional hospitals in southern China, so generalization to other regions requires caution. Fourth, due to the unavailability of cause-specific EDVs data, this study only evaluated the effects on overall EDVs. However, since EDVs caused by external factors were not excluded in the analysis of all-cause EDVs, the impact of PM may possibly have been underestimated in this study.

## 5. Conclusions

In summary, our finding provided comparative insights for an elevated risk of PEDVs related to exposure to environmental PM_1_, PM_2.5,_ and high PM_1_/PM_2.5_ ratio within a few hours. Additionally, we observed that the PM-associated risks for PEDVs were distinctly stronger in cold months than in warmer months. This study added to the evidence of hourly PM_1_ health hazards and a quantitative description of the role of PM_1_ in PM_2.5_, which could inform the development of hourly air quality standards and the optimization of emergency healthcare resources. In view of the sparse epidemiological evidence all over the world, more PM_1_-health surveys are required going forward to better characterize the impact on wellness linked to environmental fine particulate atmospheric pollution.

## Figures and Tables

**Figure 1 ijerph-20-04910-f001:**
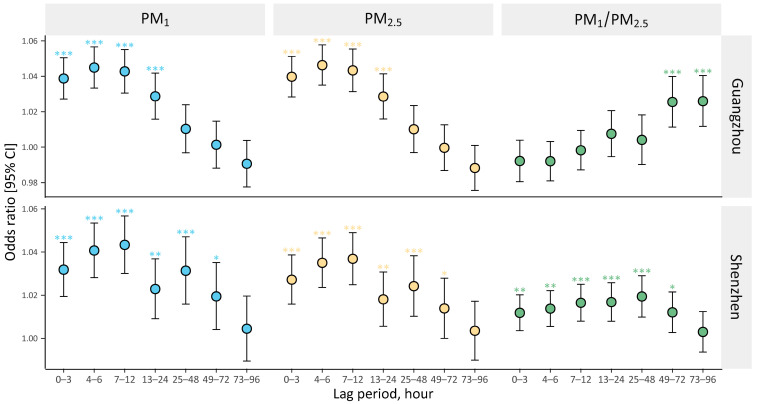
Odds ratios (with 95% CIs) for PEDVs across various exposure hours in Guangzhou and Shenzhen, associated with per IQR rise in PM_1_ and PM_2.5_, and per 10% rise in PM_1_/PM_2.5_ ratio. Notes: **p* < 0.05; ***p* < 0.01; ****p* < 0.001. Abbreviations: PEDVs, pediatric emergency department visits; PM_1_, particulate matter with aerodynamic diameter ≤ 1 μm; PM_2.5_, particulate matter with aerodynamic diameter ≤ 2.5 μm.

**Figure 2 ijerph-20-04910-f002:**
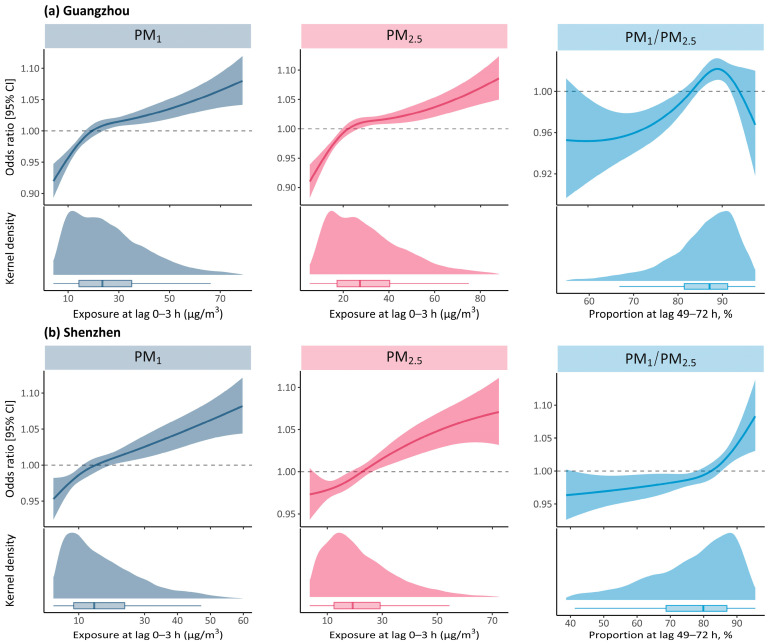
Exposure-response curves of PM_1_, PM_2.5_ and PM_1_/PM_2.5_ ratio associated with PEDVs. Abbreviations: PEDVs, pediatric emergency department visits; PM_1_, particulate matter with aerodynamic diameter ≤ 1 μm; PM_2.5_, particulate matter with aerodynamic diameter ≤ 2.5 μm.

**Figure 3 ijerph-20-04910-f003:**
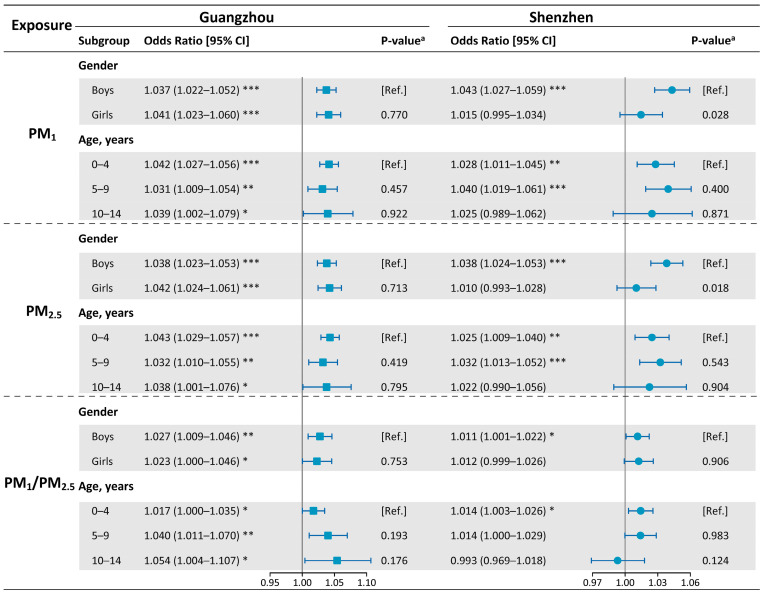
Odds ratios (with 95% CIs) for PEDVs stratified by sex and age among subgroups in Guangzhou and Shenzhen, associated with per IQR rise in PM_1_ and PM_2.5_ and per 10% rise in the PM_1_/PM_2.5_ ratio. Notes: *P*-value ^a^ indicates discrepancies between subgroups. **p* < 0.05; ***p* < 0.01; ****p* < 0.001. Abbreviations: PM_1_, particulate matter with aerodynamic diameter ≤ 1 μm; PM_2.5_, particulate matter with aerodynamic diameter ≤ 2.5 μm; PEDVs, pediatric emergency department visits.

**Table 1 ijerph-20-04910-t001:** Essential characteristics of pediatric emergency department visits in Guangzhou and Shenzhen, China, 2015–2016.

Characteristics	Guangzhou	Shenzhen
Total included visits , No.	97,508	101,639
Hourly visits, No.		
Mean (SD)	5.6 (5.5)	6.3 (4.3)
Median (IQR)	4.0 (8.0)	6.0 (6.0)
Sex, No. (%)		
Boy	58,941 (60.4)	61,889 (60.9)
Girl	38,567 (39.6)	39,750 (39.1)
Age, No. (%)		
0–4 year	64,930 (66.6)	55,680 (54.8)
5–9 year	24,046 (24.7)	34,556 (34.0)
10–14 year	8532 (8.8)	11,403 (11.2)

Abbreviations: SD, standard deviation; IQR, Interquartile range.

**Table 2 ijerph-20-04910-t002:** Hourly distributions in ambient air pollution and meteorological factors during 2015–2016 in Guangzhou and Shenzhen.

Variables	Guangzhou					Shenzhen		
	Min	Mean (SD)	Median (IQR)	Max	Min	Mean (SD)	Median (IQR)	Max
**On case hours**	*n* = 97,508				*n* = 101,639			
Particulate pollutants								
PM_1_, μg/m^3^	0.6	26.9 (16.6)	23.8 (21.6)	121.5	1.0	17.9 (13.1)	14.3 (16.0)	118.7
PM_2.5_, μg/m^3^	1.0	31.3 (18.5)	27.7 (23.7)	153.3	1.5	22.7 (14.8)	19.0 (17.2)	141.0
PM_1_/PM_2.5_, %	11.4	84.5 (8.8)	86.2 (10.7)	99.6	25.7	75.9 (14.3)	79.6 (20.3)	98.2
Gaseous pollutants								
NO_2_, μg/m^3^	4.0	52.9 (27.3)	47.0 (30.0)	226.0	6.0	33.4 (14.9)	30.0 (17.0)	125.0
SO_2_, μg/m^3^	2.0	11.6 (5.7)	10.0 (6.0)	100.0	3.0	8.0 (2.9)	7.0 (3.0)	37.0
CO, mg/m^3^	0.4	1.0 (0.3)	0.9 (0.3)	3.1	0.4	0.8 (0.2)	0.8 (0.2)	1.7
O_3_, μg/m^3^	1.0	41.4 (43.7)	25.0 (45.0)	342.0	10.0	58.3 (33.3)	49.0 (41.0)	245.0
Weather conditions								
Temperature, °C	2.5	22.8 (6.2)	24.5 (8.0)	37.5	3.0	24.7 (5.4)	26.0 (8.0)	37.0
Relative humidity, %	16.5	80.1 (14.0)	83.2 (19.3)	100.0	16.5	79.4 (12.1)	83.0 (15.2)	100.0
**On control hours**	*n* = 332,289				*n* = 347,231			
Particulate pollutants								
PM_1_, μg/m^3^	0.6	26.4 (16.4)	23.3 (21.5)	121.5	1.0	17.7 (13.0)	14.1 (16.0)	118.7
PM_2.5_, μg/m^3^	1.0	30.7 (18.2)	27.3 (23.7)	164.1	1.5	22.6 (14.7)	18.9 (16.9)	141.0
PM_1_/PM_2.5_, %	11.4	84.5 (8.9)	86.1 (10.8)	99.6	25.7	75.7 (14.4)	79.4 (20.7)	98.2
Gaseous pollutants								
NO_2_, μg/m^3^	4.0	51.9 (26.8)	47.0 (28.0)	226.0	6.0	33.2 (14.8)	30.0 (17.0)	125.0
SO_2_, μg/m^3^	2.0	11.4 (5.7)	10.0 (6.0)	100.0	3.0	7.9 (2.9)	7.0 (3.0)	37.0
CO, mg/m^3^	0.4	1.0 (0.3)	0.9 (0.3)	3.1	0.4	0.8 (0.2)	0.8 (0.2)	1.7
O_3_, μg/m^3^	1.0	40.7 (42.9)	25.0 (45.0)	342.0	10.0	57.9 (33.2)	49.0 (41.0)	245.0
Weather conditions								
Temperature, °C	2.5	22.7 (6.3)	24.5 (9.0)	37.5	3.0	24.5 (5.6)	26.0 (8.0)	37.0
Relative humidity, %	16.5	80.2 (14.1)	83.4 (19.5)	100.0	16.5	79.5 (12.3)	83.1 (15.2)	100.0

Abbreviations: SD, standard deviation; IQR, Interquartile range; NO_2_, nitrogen dioxide; SO_2_, sulfur dioxide; CO, carbon monoxide; O_3_, ozone; PM_1_, particulate matter with aerodynamic diameter ≤ 1 μm; PM_2.5_, particulate matter with aerodynamic diameter ≤ 2.5 μm.

**Table 3 ijerph-20-04910-t003:** Season-specific ORs (with 95% CIs) for PEDVs in Guangzhou and Shenzhen, associated with per IQR rise in PM_1_, PM_2.5_, and per 10% rise in PM_1_/PM_2.5_ ratio.

Exposure	Guangzhou		Shenzhen	
	OR [95% CI]	P for Heterogeneity	OR [95% CI]	P for Heterogeneity
PM_1_, lag 0–3 h		0.004		0.011
Cold	1.068 [1.053 to 1.084]		1.037 [1.021 to 1.054]	
Warm	1.037 [1.021 to 1.052]		1.010 [0.998 to 1.023]	
PM_2.5_, lag 0–3 h		0.005		0.025
Cold	1.072 [1.057 to 1.088]		1.034 [1.019 to 1.051]	
Warm	1.040 [1.025 to 1.056]		1.011 [0.998 to 1.024]	
PM_1_/PM_2.5_, lag 49–72 h		0.025		0.576
Cold	1.044 [1.009 to 1.080]		1.015 [0.991 to 1.040]	
Warm	1.000 [0.984 to 1.017]		1.008 [0.997 to 1.018]	

Abbreviations: CI, confidence interval; PEDVs, pediatric emergency department visits; PM_1_, particulate matter with aerodynamic diameter ≤ 1 μm; PM_2.5_, particulate matter with aerodynamic diameter ≤ 2.5 μm; Warm season, April to September; Cold season, October to March of the next year.

## Data Availability

Not applicable.

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
