# Peer review of "Emergency Department Visits in Children Associated with Exposure to Ambient PM1 within Several Hours"

_ijerph, 2023, doi:10.3390/ijerph20064910_

Round 1

Reviewer 1 Report

This manuscript assessed the relationship between ambient PM1 and emergency department visits for children within several hours. The topic is interest in the field of environmental epidemiology and the manuscript is generally well organized, but some additional analyses are still required. Before acceptance for publication, I had several minor suggestions when I read this manuscript.

1. Public holidays usually influence people's daily behavior and lifestyle and may lead to some differences in health outcomes (e.g., hospital admission). In the case cross-over design, the factor of public holidays should also be controlled for in the conditional logistic model. 

2. Correlations exist between pollutants, and single-pollutant models cannot reflect the effects of such associations.  It is good that the authors performed sensitivity analysis using bi-pollutant models to check the robustness of the estimated associations. I will also suggest to use multiple-pollutant models (e.g., tri-pollutant) to further account for the effects of co-pollutants..

3. There are several grammatical errors throughout the paper. I suggest that authors carefully review the english writing of paper. Specific comments:

Line 167,198,212, " correlated with" should be changed to "associated with "; Line 239, " effect pattern " should be revised as "effect patterns"; Line 203, " perceived” should be replaced by "observed".

Author Response

  1. Public holidays usually influence people's daily behavior and lifestyle and may lead to some differences in health outcomes (e.g., hospital admission). In the case cross-over design, the factor of public holidays should also be controlled for in the conditional logistic model.

Response:

We agree with the reviewer's insightful comments and updated our analyses accordingly. As suggested, we incorporated holiday indicators in the conditional logistic model to control for potential holiday effects on PEDVs. The updated estimates were overall similar with those derived from our original analysis (See table presented below). The corresponding descriptions were added in the revised manuscript and the supplementary materials were also updated accordingly.

(Lines 104–106 in the Methods):

To control for the potential holiday effects on staffing and service delivery[1], we included a dichotomous variable in the model to indicate whether a particular date was a national public holiday.

Citation:

  1. Walker, A.S.; Mason, A.; Quan, T.P.; Fawcett, N.J.; Watkinson, P.; Llewelyn, M.; Stoesser, N.; Finney, J.; Davies, J.; Wyllie, D.H.; et al. Mortality risks associated with emergency admissions during weekends and public holidays: an analysis of electronic health records. Lancet 2017, 390, 62-72, doi:10.1016/S0140-6736(17)30782-1.

Table Comparison of main results between updated and original analyses in Guangzhou.

Pollutant

Lag

OR [95% CI]

Updated results

Original results

PM1

0–3 h

1.039 (1.027–1.050)

1.038 (1.027–1.050)

4–6 h

1.045 (1.033–1.057)

1.045 (1.033–1.056)

7–12 h

1.043 (1.031–1.055)

1.043 (1.031–1.055)

13–24 h

1.029 (1.016–1.042)

1.029 (1.016–1.042)

25–48 h

1.010 (0.997–1.024)

1.011 (0.998–1.025)

49–72 h

1.001 (0.988–1.015)

1.003 (0.990–1.016)

73–96 h

0.991 (0.978–1.004)

0.993 (0.980–1.006)

PM2.5

0–3 h

1.040 (1.028–1.051)

1.039 (1.028–1.051)

4–6 h

1.046 (1.035–1.058)

1.046 (1.034–1.057)

7–12 h

1.043 (1.031–1.055)

1.043 (1.031–1.055)

13–24 h

1.029 (1.016–1.041)

1.029 (1.016–1.042)

25–48 h

1.010 (0.997–1.023)

1.011 (0.998–1.024)

49–72 h

1.000 (0.987–1.013)

1.001 (0.988–1.014)

73–96 h

0.988 (0.976–1.001)

0.991 (0.978–1.003)

PM1/PM2.5

0–3 h

0.992 (0.981–1.004)

0.994 (0.982–1.006)

4–6 h

0.992 (0.981–1.003)

0.994 (0.983–1.005)

7–12 h

0.998 (0.987–1.009)

0.999 (0.988–1.011)

13–24 h

1.008 (0.995–1.021)

1.009 (0.996–1.022)

25–48 h

1.004 (0.990–1.018)

1.006 (0.992–1.020)

49–72 h

1.026 (1.011–1.040)

1.027 (1.013–1.041)

73–96 h

1.026 (1.012–1.040)

1.028 (1.013–1.042)

Table Comparison of main results between updated and original analyses in Shenzhen.

Pollutant

Lag

OR [95% CI]

Updated results

Original results

PM1

0–3 h

1.032 (1.019–1.044)

1.032 (1.020–1.044)

4–6 h

1.041 (1.028–1.053)

1.041 (1.028–1.053)

7–12 h

1.043 (1.030–1.057)

1.043 (1.030–1.057)

13–24 h

1.023 (1.009–1.037)

1.023 (1.009–1.037)

25–48 h

1.031 (1.016–1.047)

1.031 (1.016–1.047)

49–72 h

1.019 (1.004–1.035)

1.020 (1.004–1.035)

73–96 h

1.004 (0.990–1.020)

1.004 (0.989–1.019)

PM2.5

0–3 h

1.027 (1.016–1.039)

1.027 (1.016–1.039)

4–6 h

1.035 (1.024–1.046)

1.035 (1.024–1.047)

7–12 h

1.037 (1.025–1.049)

1.037 (1.025–1.049)

13–24 h

1.018 (1.006–1.031)

1.018 (1.006–1.031)

25–48 h

1.024 (1.010–1.038)

1.024 (1.011–1.039)

49–72 h

1.014 (1.000–1.028)

1.014 (1.000–1.028)

73–96 h

1.004 (0.990–1.017)

1.003 (0.990–1.017)

PM1/PM2.5

0–3 h

1.012 (1.004–1.020)

1.012 (1.003–1.020)

4–6 h

1.014 (1.005–1.022)

1.014 (1.005–1.022)

7–12 h

1.016 (1.008–1.025)

1.016 (1.008–1.025)

13–24 h

1.017 (1.008–1.026)

1.016 (1.007–1.025)

25–48 h

1.019 (1.010–1.029)

1.019 (1.009–1.028)

49–72 h

1.012 (1.003–1.022)

1.011 (1.002–1.021)

73–96 h

1.003 (0.994–1.012)

1.002 (0.993–1.012)

  1. Correlations exist between pollutants, and single-pollutant models cannot reflect the effects of such associations. It is good that the authors performed sensitivity analysis using bi-pollutant models to check the robustness of the estimated associations. I will also suggest to use multiple-pollutant models (e.g., tri-pollutant) to further account for the effects of co-pollutants.

Response:

Thank you for your insightful comments. According to the reviewer’s suggestion, we additionally performed tri-pollutant analyses to account for the effects of co-pollutants. In order to avoid potential collinearity between pollutants in multi-pollutant models, we included only those pollutants with Spearman’s correlation coefficients <0.7. In addition, we performed variance inflation factor (VIF) analysis to test for the presence of multicollinearity in the multi-pollutant model. The analysis indicated that our analyses are probably not affected by multicollinearity since the VIF value is all less than 2. As shown in the revised Table S3 (presented below), the updated estimates changed little compared to the estimates obtained by our preliminary analysis using the single-pollutant model. All these sensitivity analyses provided further support for the robustness of the reported PM-PEDVs associations in this paper.

(Lines 133–137 in the Methods):

We used bi- and tri-pollutant models to account for the effects of co-exposures to gaseous pollutants (e.g., NO2, SO2, CO, and O3). To avoid potential multicollinearity, we only included pollutants with Spearman’s correlation coefficients <0.7. In addition, we performed a variance inflation factor (VIF) analysis to test for the presence of multicollinearity in the multi-pollutant models [2].”

Citation:

  1. Shanahan, K.H.; Subramanian, S.V.; Burdick, K.J.; Monuteaux, M.C.; Lee, L.K.; Fleegler, E.W. Association of Neighborhood Conditions and Resources for Children With Life Expectancy at Birth in the US. JAMA network open 2022, 5, e2235912, doi:10.1001/jamanetworkopen.2022.35912.

Table S3 Sensitive analysis of PM-PEDVs associations in Guangzhou and Shenzhen by changing modelling choices of conditional logistic regression.

Models

OR (95% CI) per IQR increase in PM1 and PM2.5

Guangzhou

Shenzhen

PM1

PM2.5

PM1

PM2.5

Single-pollutant 

1.045 (1.033–1.057)

1.046 (1.035–1.058)

1.041 (1.028–1.053)

1.035 (1.024–1.046)

Bi-pollutant

+ NO2

1.027 (1.012–1.042)

1.030 (1.015–1.045)

1.037 (1.023–1.051)

1.031 (1.018–1.044)

+ SO2

1.038 (1.024–1.052)

1.040 (1.026–1.054)

1.034 (1.020–1.048)

1.028 (1.016–1.041)

+ CO

1.041 (1.028–1.055)

1.043 (1.030–1.056)

1.048 (1.034–1.062)

1.040 (1.027–1.052)

+ O3

1.045 (1.032–1.057)

1.046 (1.034–1.058)

1.040 (1.027–1.054)

1.034 (1.021–1.046)

Tri-pollutant

+ NO2 + SO2

1.025 (1.009–1.041)

1.028 (1.012–1.044)

1.033 (1.019–1.048)

1.027 (1.014–1.040)

+ NO2 + CO

1.027 (1.012–1.043)

1.030 (1.015–1.045)

1.043 (1.028–1.058)

1.035 (1.022–1.048)

+ NO2 + O3

1.020 (1.004–1.036)

1.023 (1.007–1.039)

1.034 (1.018–1.049)

1.027 (1.013–1.041)

+ SO2 + CO

1.034 (1.019–1.049)

1.037 (1.022–1.051)

1.041 (1.026–1.056)

1.033 (1.020–1.046)

+ SO2 + O3

1.036 (1.021–1.050)

1.038 (1.024–1.052)

1.034 (1.020–1.048)

1.028 (1.015–1.041)

+ CO + O3

1.038 (1.023–1.052)

1.039 (1.026–1.053)

1.047 (1.033–1.062)

1.039 (1.025–1.052)

Abbreviations: CI, confidence interval; OR, odds ratio; PEDVs, pediatric emergency department visits; PM1, particulate matter with aerodynamic diameter ≤1 μm; PM2.5, particulate matter with aerodynamic diameter ≤2.5 μm; SO2, sulfur dioxide; NO2, nitrogen dioxide; CO, carbon monoxide; O3, ozone.

  1. There are several grammatical errors throughout the paper. I suggest that authors carefully review the English writing of paper. Specific comments: Line 167,198,212, " correlated with" should be changed to "associated with"; Line 239, " effect pattern " should be revised as "effect patterns"; Line 203, " perceived” should be replaced by "observed".

Response:

Great thanks for your reminder. Changes are made in our revised submission, please see our revised manuscript for more details.

Reviewer 2 Report

The article was generally written well. I have a few suggestions for the authors to revise the current manuscript.

1. Some cases are likely to be due to accidental events or infectious diseases, and air pollution exposure may have little impact on these events. Please clarify whether PEDVs are recorded for specific reasons in the study, and if the data are not available, they should be listed as a limitation.

2. Air pollutants are often correlated with each other. It is recommended to use both two- and three-pollutant models for sensitivity analyses of PM-PEDV associations.

3. Replace Figure S1 in the supplementary material with a higher resolution image.

4. Please change "P-value a" to "P-value" in Table S1 and S2, as no notes were provided. Modifications are suggested to change "P for interaction" in Table 3 and "P-value a" in Tables S4 and S5 to "P for heterogeneity".

5. Please add an abbreviation note for "df" in Table S3, which will better help the reader understand the results.

Author Response

  1. Some cases are likely to be due to accidental events or infectious diseases, and air pollution exposure may have little impact on these events. Please clarify whether PEDVs are recorded for specific reasons in the study, and if the data are not available, they should be listed as a limitation.

Response:

Thanks a lot for your suggestions. Due to the lack of cause-specific data, we were unable to investigate the associations between size-specific PMs and cause-specific EDVs in this study. Similarly, we could not exclude EDVs due to accidents (e.g., road traffic injuries and fires). Therefore, we may have underestimated the impact of PMs on hourly EDVs to some extent [3]. As the data is not available, we acknowledge this as a limitation of our study.

(Lines 274–277):

Fourth, due to the unavailability of cause-specific EDVs data, this study only evaluated the effects on overall EDVs. However, since EDVs caused by external factors were not excluded in the analysis of all-cause EDVs, the impact of PM may possibly have been underestimated in this study.

Citation:

  1. Cao, J.; Li, W.; Tan, J.; Song, W.; Xu, X.; Jiang, C.; Chen, G.; Chen, R.; Ma, W.; Chen, B.; et al. Association of ambient air pollution with hospital outpatient and emergency room visits in Shanghai, China. The Science of the total environment 2009, 407, 5531-5536, doi:10.1016/j.scitotenv.2009.07.021.
  2. Air pollutants are often correlated with each other. It is recommended to use both two- and three-pollutant models for sensitivity analyses of PM-PEDV associations.

Response:

Thanks for your comments. According to the suggestion, we added sensitivity analysis using three-pollutant models in the revised Supplementary Material to evaluate the robustness of our results. The updated estimates are approximately comparable with those obtained from the single-pollutant analyses. Please refer to Table S3 in comment 2 to reviewer 1.

  1. Replace Figure S1 in the supplementary material with a higher resolution image.

Response:

Thanks for your insightful comment. According to your suggestion, I have replaced Figure S1 with a higher resolution image.

For details, see the annex.

Fig. S1. Spearman correlation matrix between ambient air pollutants and meteorological factors in Guangzhou and Shenzhen, China, 2015–2016. Abbreviations: PM1, particulate matter with aerodynamic diameter ≤1 μm; PM2.5, particulate matter with aerodynamic diameter ≤2.5 μm; NO2, nitrogen dioxide; SO2, sulfur dioxide; O3, ozone; CO, carbon monoxide; Temp, temperature; RH, relative humidity.

4.Please change "P-valuea" to "P-value" in Table S1 and S2, as no notes were provided. Modifications are suggested to change "P for interaction" in Table 3 and "P-valuea" in Tables S4 and S5 to "P for heterogeneity".

Response:

According to the reviewer comments, we have made changes to the corresponding places. Our revised Table 3 is shown below and the reviewer can see the revised supplementary materials for more details of Tables S1, S2, S4 and S5.

Table 3 Season-specific ORs (with 95% CIs) for PEDVs in Guangzhou and Shenzhen, associated with per IQR rise in PM1, PM2.5, and per 10% rise in PM1/PM2.5.

Exposure

Guangzhou

Shenzhen

OR [95% CI]

P for heterogeneity

OR [95% CI]

P for heterogeneity

PM1, lag 0–3 h

0.004

0.011

Cold

1.068 [1.053 to 1.084]

1.037 [1.021 to 1.054]

Warm

1.037 [1.021 to 1.052]

1.010 [0.998 to 1.023]

PM2.5, lag 0–3 h

0.005

0.025

Cold

1.072 [1.057 to 1.088]

1.034 [1.019 to 1.051]

Warm

1.040 [1.025 to 1.056]

1.011 [0.998 to 1.024]

PM1/PM2.5, lag 49–72 h

0.025

0.576

Cold

1.044 [1.009 to 1.080]

1.015 [0.991 to 1.040]

Warm

1.000 [0.984 to 1.017]

1.008 [0.997 to 1.018]

Abbreviations: CI, confidence interval; PEDVs, pediatric emergency department visits; PM1, particulate matter with aerodynamic diameter ≤1 μm; PM2.5, particulate matter with aerodynamic diameter ≤2.5 μm; Warm season, April to September; Cold season, October to March of the next year.

  1. Please add an abbreviation note for "df" in Table S3, which will better help the reader understand the results.

Response:

Thanks for your suggestion. We have added an abbreviation note for "df" in Table S3. Please see the revised Supplementary Materials for details.

(Lines 52–54):

Abbreviations: CI, confidence interval; OR, odds ratio; PEDVs, pediatric emergency department visits; PM1, particulate matter with aerodynamic diameter ≤1 μm; PM2.5, particulate matter with aerodynamic diameter ≤2.5 μm; SO2, sulfur dioxide; NO2, nitrogen dioxide; CO, carbon monoxide; O3, ozone; Temp, temperature; RH, relative humidity; df, degree of freedom.

Reviewer 3 Report

It would be very interesting for readers if a list of acronyms and abbreviations were inserted into the manuscript.

Author Response

Response:

We greatly appreciate the reviewer’s comments for our manuscript. As suggested, we have added the list of abbreviations for clarity.

Abbreviations   

Terminology

PM

particulate matter

PM0.1

particulate matter with aerodynamic diameter ≤0.1 μm

PM1

particulate matter with aerodynamic diameter ≤1 μm

PM2.5

particulate matter with aerodynamic diameter ≤2.5 μm

NO2

nitrogen dioxide

SO2

sulfur dioxide

CO

carbon monoxide

O3

ozone

EDVs

emergency department visits

PEDVs

pediatric emergency department visits

CI

confidence interval

OR

odds ratio

TSCC

time-stratified case-crossover design

CLR

conditional logistic regression

NCS

natural cubic spline

df

degree of freedom

IQR

interquartile range

MR

meta regression

SD

standard deviation
